PERSPECTIVE

# Guided by Microbes: Applying Community Coalescence Principles for Predictive Microbiome Engineering

Jennifer D. Rocca,[a] Mario E. Muscarella,[b] Ariane L. Peralta,[c] Dandan Izabel-Shen,[d] Marie Simonin[e]

aDepartment of Plant and Microbial Biology, North Carolina State University, Raleigh, North Carolina, USA
bInstitute of Arctic Biology, Department of Biology and Wildlife, University of Alaska Fairbanks, Fairbanks, Alaska, USA
cDepartment of Biology, East Carolina University, Greenville, North Carolina, USA
dDepartment of Ecology, Environment, and Plant Sciences, Stockholm University, Stockholm, Sweden
eUniversity of Angers, Institut Agro, INRAE, IRHS, SFR QUASAV, Angers, France

**ABSTRACT** Every seed germinating in soils, wastewater treatment, and stream confluence exemplify microbial community coalescence—the blending of previously isolated communities. Here, we present theoretical and experimental knowledge on how separated microbial communities mix, with particular focus on managed ecosystems. We adopt the community coalescence framework, which integrates metacommunity theory and meta-ecosystem dynamics, and highlight the prevalence of these coalescence events within microbial systems. Specifically, we (i) describe fundamental types of community coalescences using naturally occurring and managed examples, (ii) offer ways forward to leverage community coalescence in managed systems, and (iii) emphasize the importance of microbial ecological theory to achieving desired coalescence outcomes. Further, considering the massive dispersal events of microbiomes and their coalescences is pivotal to better predict microbial community dynamics and responses to disturbances. We conclude our piece by highlighting some challenges and unanswered question yet to be tackled.

**KEYWORDS** biostimulants, community coalescence, community inoculants, managed ecosystems, microbiome engineering

## COMMUNITY COALESCENCE: DEFINITION, OCCURRENCES, AND KNOWLEDGE SYNTHESIS FROM CASE-STUDIES

Community coalescence—dispersal en masse—is a pervasive process in the microbial realm (1–4). For example, leaf and soil microbiomes make contact and integrate during litterfall (5). Blending separated communities often occurs in tandem with their respective environments, characteristic of stream confluences (2). Community coalescence can be ephemeral (e.g., monsoonal erosion) or perpetual (e.g., tidal ebb/flow), with restricted spatial distribution (e.g., bird fecal deposits) or be widespread (soil blowing across the landscape). These naturally occurring coalescences may be important components of ecosystem function but are also key features of managed systems where the coalescences are intentionally implemented or controlled (1, 6).

Microbial engineering historically focused on isolating and imposing directed artificial selection, or selective enrichment, on one or a few microorganisms, to optimize the strain(s) for a particular function or set of functions. This approach largely overlooks the biotic interactions with the receiving community for the success/failure of these strains (7–9) (Fig. 1a). Embracing a top-down approach, Swenson et al. introduced the concept of "ecosystem selection" (10), wherein whole ecosystems, comprising many species and their interactions, are targeted and tested for optimizing engineered systems, without the colossal task of examining every species or all possible interactions. A recent study demonstrates that applying repeated perturbations on microbial communities

Address correspondence to Jennifer D. Rocca, jenny.rocca@gmail.com.

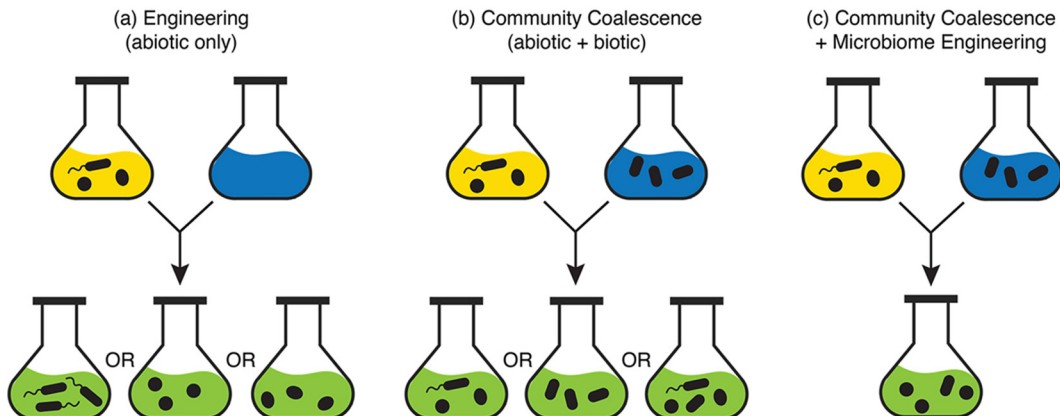

**FIG 1** Conceptual representation of intentional microbiome management. The three panels depict how (a) engineering with regard to manipulating abiotic conditions only or (b) mixing two previously isolated communities and associated environments can result in different community outcomes; in contrast (c), combining approaches of intentional microbiome engineering of specific synthetic communities and considering/addressing how environmental conditions affect resultant community assembly refines the predictability of microbiome management.

as an artificial selection strategy results in microbial consortia with high functionality and invasion resistance (11). As such, using wholescale mixing of isolated communities in designing microbial consortia (Fig. 1b) should enable more predictive outcomes for microbiome engineering (12), agriculture (13), and wastewater systems (6) (Fig. 1c). Here, we expand on this concept by exemplifying the utility of community coalescence pertinent to managed systems and offer insight into how aspects of community ecology should inform the engineering of microbial communities.

We start by introducing one of the most important managed systems—a municipal effluent and wastewater treatment plant (WWTP), where coalescing of communities occurs consistently and smoothly through incidental and intentional merging (Fig. 2a). Entry into the WWTP unidirectionally coalesces eutrophic sewage into a distinct system with acute abiotic disturbances and optimized microbial consortia to reduce the nutrient, microbial, and toxin loads of the effluent (14–16). In other environmental contexts, similar convergence of drastically different microbial communities also occurs with the bidirectional movement of marine waters into freshwater wetlands along every coastal margin, or where leaky mine tailings unidirectionally flow into stream networks. The merged WWTP effluent continues through a series of engineered abiotic disturbances (Fig. 2a) (17), where supplemental microbial consortia are added to optimize nitrification and denitrification in order to mitigate the nutrient load (18). The supplemented microbial consortia in the WWTP are effectively repeated community coalescences, recycled to amend municipal effluent. Like a sourdough bread starter, the consortia are added to the system to acquire a desired function, then subset and retained to repeat the process. As demonstrated *in silico* by Chang et al. (11), imposing repeated community coalescence perturbations on microbial communities can be an effective strategy of artificial community selection for developing high-performing microbial consortia.

Ideally, the reintroduction of the now cleaned WWTP water into the natural environment comprises oligotrophic water and a low-diversity microbial community (Fig. 2a) (18). In other instances, the managed microbiome may instead be optimized for high diversity and to withstand reassembly into the environment. For example, microbial consortia added to agricultural systems for enhanced productivity or reduced pathogen load are intended to persevere in the merged community (19). Reentry of the clean WWTP effluent into natural systems should, however, not persist. Natural coalescences, such as saltwater intrusion, show that the community with higher initial diversity does not necessarily prevail (20–22). Consequently, clean effluent with a very low-diversity microbiome does not unquestionably indicate a successful cleanup and reentry; a comprehensive understanding of the particular system is crucial for predicting

mSystems®

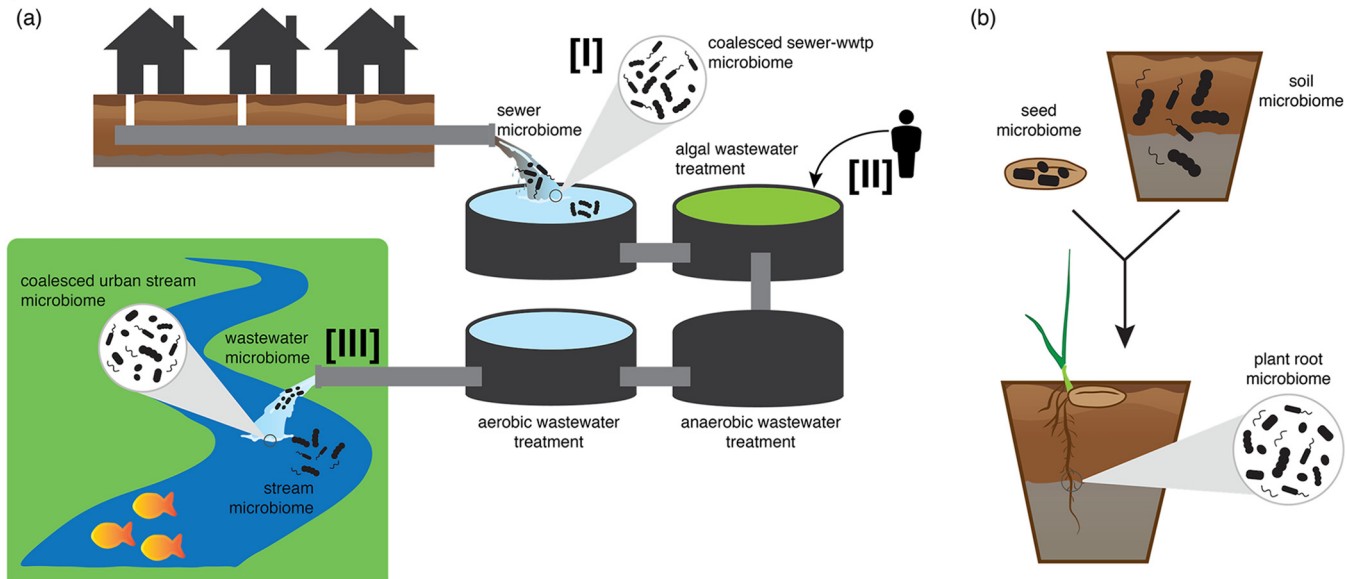

**FIG 2** Examples of community coalescences in managed systems in (a) wastewater and (b) seed-soil systems. We use the wastewater system (WWTP) in panel a to illustrate various forms of managed coalescence: (I) redirected coalescence, where municipal effluent occurs regardless, but allowing raw sewage to coalesce directly with urban and natural areas is not optimal (41, 42), so WWTPs redirect the coalescence of raw sewage, itself a mixture from urban infrastructure, for preprocessing; (II) intentional application of microbial consortia, which constitutes an engineered microbiome able to withstand repeated coalescence exposure while maintaining desired community function; and (III) mitigating the release of microbiomes back into nature. In panel b, we illustrate coalescence in a host-associated context with the distinct microbiome of a plant seed interacting with the resident soil microbial community to result in the plant root (i.e., rhizosphere) microbiome, where the importance of rare taxon emergence and pathogen inhibition are optimal criteria for engineered seed microbiomes.

what communities will prevail or subside once coalesced with other communities. These pioneer studies indicate that microbial coalescences are often characterized by "surprising" outcomes where dominant taxa are replaced by initially rare ones or where a community of high biomass and diversity is readily replaced by a low-biomass-diversity counterpart. Such observations indicate that coalescences that are extremely pervasive in natural or managed systems present opportunities to understand the predictability of the system or even intervention points to manage microbial communities for a given outcome.

## LEVERAGING COALESCENCES FOR EFFICIENT MICROBIOME ENGINEERING

The previous examples demonstrate that microbial community coalescence represents a massive biotic perturbation capable of modifying all the attributes of microbiomes, including diversity, composition, function, or resilience to disturbances. As we improve our capacity to predict the outcomes of community coalescence in various environmental contexts, it becomes more evident that this framework offers tremendous possibilities to engineer microbial communities to energize or produce targeted functions.

Community coalescence can be harnessed to construct a target microbial community, comprising (i) robust inoculants with a high degree of compositional and functional resistance/resilience to changing conditions (e.g., wastewater treatment plant, agricultural soils) or intentionally not persistent, as would be the ideal target for the WWTP clean outflow, (ii) communities occupying a specific niche or with competitive capacities to prevent (e.g., pests or pathogens) or stimulate (e.g., beneficial or keystone organisms) the presence of specific taxa, (iii) the optimization of microbial consortia performing specific microbial functions (e.g., biogas production from organic waste, ruminal fermentation in animals), and (iv) the restoration of communities following important biotic or abiotic disturbances (e.g., soil remediation, antibiotic treatment, disease).

Performing entire microbial community mixing with or without their source environment may prove more efficient than using the inoculation of a single microbial strain (10). Single-strain inoculation has faced many challenges hampering its success

in the past decade, mainly due to a lack of persistence in the target environment leading to a low efficacy and reapplication costs (23, 24). Several examples of successful microbiome engineering, based on community coalescence, offer some degree of assurance in the reliability of the approach. In the human health context, the use of fecal microbiota transplantation to treat several conditions (e.g., *Clostridium difficile* infection, ulcerative colitis, antibiotic-treated patients) demonstrates the efficiency of the approach to restore microbiomes by outcompeting unwanted taxa (25–27). Other examples with good reliability include soil restoration by mixing complex soil inocula or amendments, highlighting the efficacy of community blending to increase soil fertility via boosted microbial activity, increased biodiversity, and suppressed soilborne pathogens (Fig. 2b) (28–31).

The integration of ecological theory and engineering design principles into microbiome engineering remains challenging, given the complex interplay between environmental factors and biotic interactions (32). Metacommunity theory has proposed mechanisms linking the spatial dynamics of local and regional species pools, particularly the performance or fitness of local and immigrant species in the receiving environments (33). The community coalescence framework builds on earlier models of immigration dynamics by considering a distinct set of assembly processes that involves fusing whole communities along with the respective environments (1). Although this framework was developed to provide a mechanistic understanding of community assembly processes in natural systems, recent work highlights its utility to the ecological principles of optimizing community coalescence in engineering-based management (4, 34). For example, the harnessing of environmental filtering (environments select certain species) and priority effects (early colonizer impacts on the success of subsequent arrivals) (35) can inform the design and construction of target microbial consortia. Preconditioned microbial assemblages may be able to colonize newly blended communities successfully and rapidly to outperform species with undesired functions. This scenario is particularly useful for the maintenance and optimal performance of a target microbiome highly optimized for removing contaminants in wastewater treatment systems during the short water-retention period. Microbial responses to a coalescence event can be modulated by the history and environmental setting of the community (36). Preexposure of populations to a defined and coalesced environment enhances their resistance and recovery capacities when they are confronted with subsequent perturbations (37). In an environment or managed system exposed to perturbations (the change in the environment that resulted in coalescence), species with preexposure are likely to spread genes involved in the tolerance of those perturbations between organisms via horizontal gene transfer and thereby create genetic "memory" in the newly assembled communities. It is therefore important to consider the history of a source community when designing robust consortia for use in microbiome engineering. Communities subjected to repeated coalescence events may possess desired functions through the repeated biotic disturbance but be highly resistant to subsequent invasions as a potential "side effect" (11). In fact, Chang et al. (11) found that repeated perturbations of coalescence render the microbial communities more resilient to subsequent invasions. This emergence of resilience and resistance may or may not be a desired trait of the engineered consortia. In this regard, understanding how consortia were previously selected and how they respond to subsequent community merging events is crucial information for strategies of titrating the optimal resistance or resilience of the engineered community.

While great advances have been made in understanding microbial community coalescence and applying the theoretical framework in both natural and engineered contexts, many hurdles remain, and it is still difficult to predict outcomes. Community coalescence events often promote new abiotic and biotic conditions where interactions between species with unknown traits occur. Moreover, some degree of stochasticity in community assembly, including priority effects and drift (38), makes the reproducibility of community coalescence difficult to predict in engineered systems. Likewise, the

roles of higher-order interactions (32, 39, 40) and trophic interactions (e.g., viral predation) are still largely unknown and therefore make predicting microbiome dynamics challenging. How do we parameterize the collective interactions and account for stochastic events in engineered systems that require an intentional outcome? What are the unknown physiological traits that mediate successful establishment postcoalescence? Despite these challenges, the move from single species to whole community in microbiome engineering clearly demonstrates that systems can be engineered to be robust and resilient (10), with plenty of work yet to be done. The advancement of the community coalescence framework will benefit the design of effective microbiomes by addressing these open questions and will also provide novel insight and interpretations of community dynamics and consequences for ecosystem function.

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
