## [Reviewer comments · mSystems]

Guided by microbes: applying community coalescence principles for predictive microbiome engineering

Jennifer Rocca, Mario Muscarella, Ariane Peralta, Dandan Izabel-Shen, and Marie SIMONIN

Corresponding Author(s): Jennifer Rocca, North Carolina State University

Review Timeline:

Submission Date:	April 29, 2021
Editorial Decision:	May 21, 2021
Revision Received:	June 28, 2021
Accepted:	July 12, 2021

Editor: Ashley Shade

Reviewer(s): Disclosure of reviewer identity is with reference to reviewer comments included in decision letter(s). The following individuals involved in review of your submission have agreed to reveal their identity: Juan Diaz-Colunga (Reviewer #1)

Transaction Report:

DOI: <https://doi.org/10.1128/mSystems.00538-21>

May 21, 2021

Dr. Jennifer Rocca
North Carolina State University
Plant and Microbial Biology
Raleigh 27607

Re: mSystems00538-21 (Guided by microbes: applying community coalescence principles for predictive microbiome engineering)

Dear Dr. Jennifer Rocca:

Thank you for submitting your manuscript to mSystems. We have completed our review and I am pleased to inform you that, in principle, we expect to accept it for publication in mSystems. However, acceptance will not be final until you have adequately addressed the reviewer comments.

Thank you for the privilege of reviewing your work. Below you will find instructions from the mSystemseitorial office and comments generated during the review.

Preparing Revision Guidelines

For complete guidelines on revision requirements, please see the Instructions to Authors at <https://msystems.asm.org/sites/default/files/additional-assets/mSys-ITA.pdf>. **Submissions of a paper that does not conform to mSystems guidelines will delay acceptance of your manuscript.**

Sincerely,

Ashley Shade

Editor, mSystems

Journals Department
Reviewer comments:

Reviewer #1 (Comments for the Author):

In this manuscript, the authors propose microbial community coalescence as a tool to engineer microbiomes for specific functions. They argue that the success of traditional artificial selection protocols designed to improve a trait of a single organism might be hampered by the fact that biotic interactions are important determinants of the organism's viability. Instead, engineering whole microbiomes that can maximize the desired function when being mixed with the receiving community could lead to more defined outcomes. To that end, community coalescence could serve to build towards microbial communities with improved functions. They propose wastewater treatment plants (WWTPs) as a model to explore the utility of coalescence for community engineering.

The authors discuss an exciting idea. They focus on what is essentially artificial selection at the community level, i.e. the improvement of a function that is carried out by a community (as opposed to a single microorganism) through iterative protocols that favor the propagation of higher functioning consortia. Although this concept was proposed fairly long ago (e.g. Swenson et al. 2000, PNAS), to this day only very moderate success has been achieved. One of the reasons might be that artificial community selection strategies have tended to mimic those used previously for small populations of sexually reproducing animals. In that sense, coalescence as a phenomenon is specific to microbial communities, so exploiting it in that context might lead to higher success. This idea is undoubtedly novel and intriguing.

Some suggestions for the authors to polish their manuscript follow:

- 1.- I strongly suggest the authors read and discuss a recent paper by Chang et al. entitled "Engineering complex communities by directed evolution" (Nature Ecology & Evolution, 2021). In that work, Chang et al. argue that generating communities with high functionality can be achieved through serial cycles of perturbation-stabilization of previous high functioning consortia. In fact, one of the methods they propose (and test in silico) to perturb communities and potentially generate higher functioning variants is community coalescence. This is the process that takes place in

WWTPs as described in this manuscript (lines 87-88, "the consortia are added to the system to acquire a desired function, then subset and retained to repeat the process"). I think it is important to highlight that this has been tested *in silico* with promising results.

2.- The authors propose that community coalescence can be used to guide microbial communities towards states with desired properties. I agree with this in general, but one needs to be careful with the interplay between "what is being selected for" and "how is it being selected for". Specifically, in lines 113-116 the authors suggest that coalescence could be used to engineer communities that are not persistent when mixed with the receiving community (i.e. when the wastewaters are released to the environment). However, Chang et al. have shown that communities assembled through serial cycles of coalescence-selection end up being highly resilient to further invasions -- even if the function under selection is not resilience *per se*. Intuitively, one could think of this as the "what" and the "how" not being completely orthogonal (at least in reasonable selection protocols). The emergence of resilience is somewhat a "side effect" of coalescence, since communities that end up being selected are those that consistently have a high function through generations, i.e. are able to resist drops in function because they are able to resist further invasions. It would be nice for the authors to discuss in the manuscript whether they think coalescence could be ineffective for microbial community engineering in certain scenarios like the one described above.

3.- This is a very minor comment, but I feel like the authors could be more specific regarding the reasons why they introduce the WWTP system in this context. In lines 63-65 they write: "we start by introducing a municipal effluent and wastewater treatment system to serve as a handy tool to characterize the range of coalescence types...". The WWTP is not only a convenient example of a setup where different types of coalescence take place. It is also (and perhaps most importantly) a system where engineering (artificial selection) of coalesced communities can occur consistently and smoothly. I think emphasizing this would set a better ground for the following discussion.

Thank you, Reviewer #1, for the helpful feedback. Our perspective piece was made stronger with the addition of your valuable suggestions. Our responses to your feedback are in blue below and the changes in the text are highlighted in yellow:

Reviewer #1 (Comments for the Author):

In this manuscript, the authors propose microbial community coalescence as a tool to engineer microbiomes for specific functions. They argue that the success of traditional artificial selection protocols designed to improve a trait of a single organism might be hampered by the fact that biotic interactions are important determinants of the organism's viability. Instead, engineering whole microbiomes that can maximize the desired function when being mixed with the receiving community could lead to more defined outcomes. To that end, community coalescence could serve to build towards microbial communities with improved functions. They propose wastewater treatment plants (WWTPs) as a model to explore the utility of coalescence for community engineering.

The authors discuss an exciting idea. They focus on what is essentially artificial selection at the community level, i.e. the improvement of a function that is carried out by a community (as opposed to a single microorganism) through iterative protocols that favor the propagation of higher functioning consortia. Although this concept was proposed fairly long ago (e.g. Swenson et al. 2000, PNAS), to this day only very moderate success has been achieved. One of the reasons might be that artificial community selection strategies have tended to mimic those used previously for small populations of sexually reproducing animals. In that sense, coalescence as a phenomenon is specific to microbial communities, so exploiting it in that context might lead to higher success. This idea is undoubtedly novel and intriguing.

Thank you for the suggested addition of Swenson et al, which is indeed an early example of proposing whole-community engineered systems and the idea of 'community-level' selection. While only occasionally successful thus far, we agree that there might be more promise with engineered communities in the microbial realm, given their rapid growth rate and immense genetic and functional capacity. We have now cited this seminal paper throughout the text (Lines 44-48, 109-110, 165-167). We also now list the caveat of how more work needs to be done (L154-158)

Some suggestions for the authors to polish their manuscript follow:

1.- I strongly suggest the authors read and discuss a recent paper by Chang et al. entitled "Engineering complex communities by directed evolution" (Nature Ecology & Evolution, 2021). In that work, Chang et al. argue that generating communities with high functionality can be achieved through serial cycles of perturbation-stabilization of previous high functioning consortia. In fact, one of the methods they propose (and test in silico) to perturb communities and potentially generate higher functioning variants is community coalescence. This is the process that takes place in WWTPs as described in this manuscript (lines 87-88, "the consortia are added to the system to acquire a desired function, then subset and retained to repeat the process"). I think it is important to highlight that this has been tested in silico with promising results.

We appreciate directing our attention to this new study from Chang et al. We discuss the potential importance of historical coalescence on subsequent outcomes, but it is nice to see this formalized and tested with microbial meta-communities. We have addressed this throughout the text (Lines 49-51, 70-73, 146-153).

2.- The authors propose that community coalescence can be used to guide microbial communities towards states with desired properties. I agree with this in general, but one needs to be careful with the interplay between "what is being selected for" and "how is it being selected for". Specifically, in lines 113-116 the authors suggest that coalescence could be used to engineer communities that are not persistent when mixed with the receiving community (i.e. when the wastewaters are released to the environment). However, Chang et al. have shown that communities assembled through serial cycles of coalescence-selection end up being highly resilient to further invasions --even if the function under selection is not resilience per se. Intuitively, one could think of this as the "what" and the "how" not being completely orthogonal (at least in reasonable selection protocols). The emergence of resilience is somewhat a "side effect" of coalescence, since communities that end up being selected are those that consistently have a high function through generations, i.e. are able to resist drops in function because they are able to resist further invasions. It would be nice for the authors to discuss in the manuscript whether they think coalescence could be ineffective for microbial community engineering in certain scenarios like the one described above.

Thank you for this important comment. We have now added a paragraph to discuss these points on "what" and "how" consortia are selected for and to introduce unintentional side-effects of community engineering using coalescence. We highlight this example related to the emergence of community resilience to further invasions from the Chang et al. 2021 study (Lines 145-150).

3.- This is a very minor comment, but I feel like the authors could be more specific regarding the reasons why they introduce the WWTP system in this context. In lines 63-65 they write: "we start by introducing a municipal effluent and wastewater treatment system to serve as a handy tool to characterize the range of coalescence types...". The WWTP is not only a convenient example of a setup where different types of coalescence take place. It is also (and perhaps most importantly) a system where engineering (artificial selection) of coalesced communities can occur consistently and smoothly. I think emphasizing this would set a better ground for the following discussion.

Introducing WWTP as just an example for the purposes of highlighting different types of coalescence was not our intention, so we certainly appreciate the feedback here. Certainly, WWTP are interesting examples of where all 'flavors' of community merging coincidentally occur, but are also immensely important managed systems where the coalescences must operate effectively, otherwise the impacts to the surrounding environment are detrimental. We have edited the text to reflect this change (Lines 57-61).

July 12, 2021

Dr. Jennifer Rocca
North Carolina State University
Plant and Microbial Biology
Raleigh 27607

Re: mSystems00538-21R1 (Guided by microbes: applying community coalescence principles for predictive microbiome engineering)

Dear Dr. Jennifer Rocca:

Your manuscript has been accepted, and I am forwarding it to the ASM Journals Department for publication. For your reference, ASM Journals' address is given below. Before it can be scheduled for publication, your manuscript will be checked by the mSystems senior production editor, Ellie Ghatineh, to make sure that all elements meet the technical requirements for publication. She will contact you if anything needs to be revised before copyediting and production can begin. Otherwise, you will be notified when your proofs are ready to be viewed.

As an open-access publication, mSystems receives no financial support from paid subscriptions and depends on authors' prompt payment of publication fees as soon as their articles are accepted. =

Publication Fees:

We recognize that the video files can become quite large, and so to avoid quality loss ASM suggests sending the video file via <https://www.wetransfer.com/>. When you have a final version of the video and the still ready to share, please send it to Ellie Ghatineh at eghatineh@asmusa.org.

Sincerely,

Ashley Shade
Editor, mSystems

Journals Department
Phone: 1-202-942-9338